# Limiting habenular hyperactivity ameliorates maternal separation-driven depressive-like symptoms

Anna Tchenio[1,2,3,4], Salvatore Lecca[1,2,3,4], Kristina Valentinova[1,2,3,4] & Manuel Mameli[1,2,3,4]

Early-life stress, including maternal separation (MS), increases the vulnerability to develop mood disorders later in life, but the underlying mechanisms remain elusive. We report that MS promotes depressive-like symptoms in mice at a mature stage of life. Along with this behavioral phenotype, MS drives reduction of GABA$_B$-GIRK signaling and the subsequent lateral habenula (LHb) hyperexcitability—an anatomical substrate devoted to aversive encoding. Attenuating LHb hyperactivity using chemogenetic tools and deep-brain stimulation ameliorates MS depressive-like symptoms. This provides insights on mechanisms and strategies to alleviate stress-dependent affective behaviors.

[1] Institut du Fer à Moulin, Paris 75005, France. [2] Inserm, UMR-S 839, Paris 75005, France. [3] Université Pierre et Marie Curie, Paris 75005, France. [4] Department of Fundamental Neuroscience, The University of Lausanne, Lausanne 1005, Switzerland. Correspondence and requests for materials should be addressed to M.M. (email: manuel.mameli@unil.ch)

C hildhood neglect (i.e., maternal separation, MS) is aversive, has negative long-term repercussions on child development and primes depression in adulthood[1, 2]. The prolonged separation of newborn rodents from their mother represents an animal model of severe early-life stress that recapitulates aspects of child neglect. Animals undergoing MS present deficits in coping with stressful events. Moreover, such paradigm promotes anxiety, addictive and depressive-like behavioral phenotypes[3–5]. This raises the hypothesis that MS-driven behavioral adaptations may, at least partly, emerge from the dysfunction of neural circuits devoted to aversion processing.

The lateral habenula (LHb) contributes to encode aversion and negative reward prediction error as neurons in this structure are excited by external aversive stimuli[6,7]. The LHb provides such aversive-related information to monoaminergic centers suggesting a relevant role in motivated behaviors[8–10]. Stressors of different nature increase the activity of LHb neurons, and lesioning experiments indicate that the LHb regulates coping behaviors when facing aversive stimuli[11–13]. Furthermore, when such stressful events (i.e., foot-shocks) become inescapable and persistent, this produces aberrant LHb hyperactivity, which is instrumental for the emergence of depressive-like symptoms[11, 12, 14, 15].

However, whether chronic, painless stressful events, such as MS, promote depressive-like phenotypes through LHb adaptations remains elusive. If this holds true, we propose that reversal strategies restoring LHb function can ameliorate MS-dependent depressive-like symptoms.

Here, we demonstrate that, at a mature stage of life, mice undergoing MS present depressive-like behaviors along with increased LHb neuronal excitability. In addition, we show that MS-mediated reduction in GABA$_B$-GIRKs signaling is causal for such LHb hyperexcitability, and that limiting LHb neuronal activity through chemogenetics or deep brain stimulation ameliorates MS-driven behavioral phenotypes. These findings support the notion that aberrant habenular neuronal hyperactivity represents a neurobiological substrate underlying discrete stress-driven (i.e., acute and chronic) depressive-like symptoms.

## Results

**MS drives depressive-like states and habenular hyperexcitability.** We examined the early-life stress-dependent behavioral ramifications exposing mice to maternal separation (MS)[16] (Methods, Fig. 1a). Briefly, pups aged 7 days were, or not (Control), removed from their litter, separated from their mother, and kept in isolation for 6 h daily for a week. When tested three weeks later, Control and MS mice had comparable weight, locomotion and performance in the open field (Supplementary Fig. 1a–c). In contrast, MS mice presented higher failure rates when challenged with escapable foot-shocks (shuttle box), behavioral despair (higher immobility in tail suspension test; TST), and diminished sucrose preference (Fig. 1b–d). Furthermore, ex-vivo recordings in acute LHb-containing brain slices revealed that LHb neurons from MS mice exhibit hyperexcitability, with no alterations of the resting membrane potential (Fig. 1e; Supplementary Fig. 1d). Altogether, these findings suggest that MS promotes depressive-like symptoms and LHb hyperactivity later in life.

**Cellular mechanisms underlying MS-driven plasticity.** Tonic GABA$_B$-GIRK signaling controls LHb neuronal firing and its reduction promotes cell hyperexcitability[11]. Patch-clamp recordings from LHb neurons revealed that bath application of the GABA$_B$-R agonist baclofen evoke an outward current

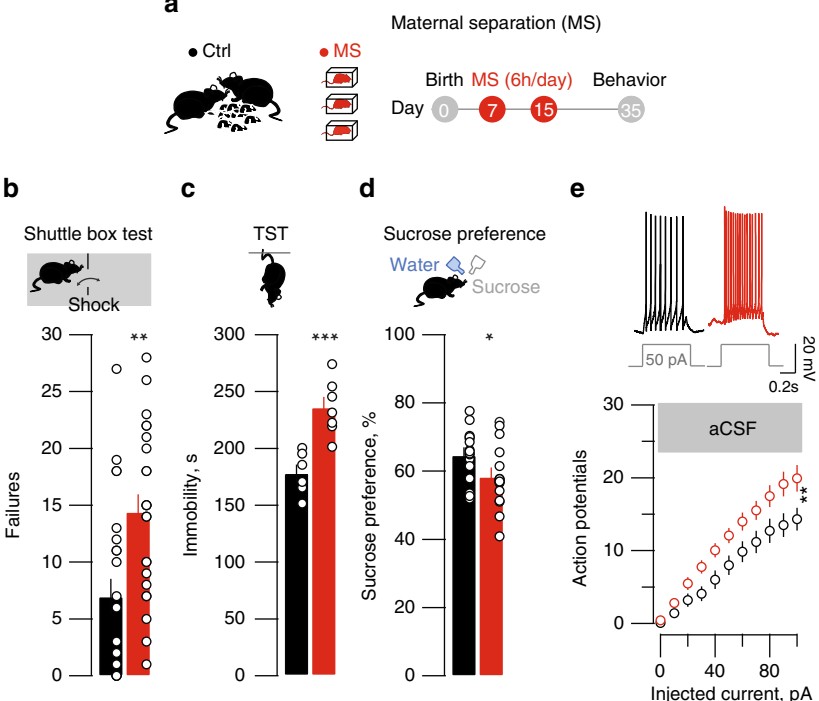

**Fig. 1** MS-induced depressive-like symptoms and hyperexcitability. **a** MS protocol (all schematics are original drawing made by authors). **b** Bar graph and scatter plot of failures in the shuttle box for control (Ctrl) and MS mice (Ctrl vs. MS; $n_{mice} = 24$, unpaired $t$-test, $t_{45} = 3.36$ **$p < 0.01$). **c** Same as **b** for TST immobility (Ctrl vs. MS; $n_{mice} = 6$–7; unpaired $t$-test, $t_{11} = 4.63$; ***$p < 0.001$). **d** Same as **b** but for sucrose preference (Ctrl vs. MS; $n_{mice} = 16$; unpaired t-test, $t_{30} = 32.12$; *$p < 0.05$). **e** Top, sample traces from Ctrl and MS mice of current-evoked firing (50 pA step). Bottom, action potentials vs. injected current (0–100 pA, steps of 10 pA) in all experimental groups (Ctrl vs. MS; aCSF, nmice = 6 per group; ncells = 23 per group; two-way ANOVA-RM, interaction, $F_{(10;440)} = 2.74$; **$p < 0.01$). Scale bars=0.2 s and 20 mV

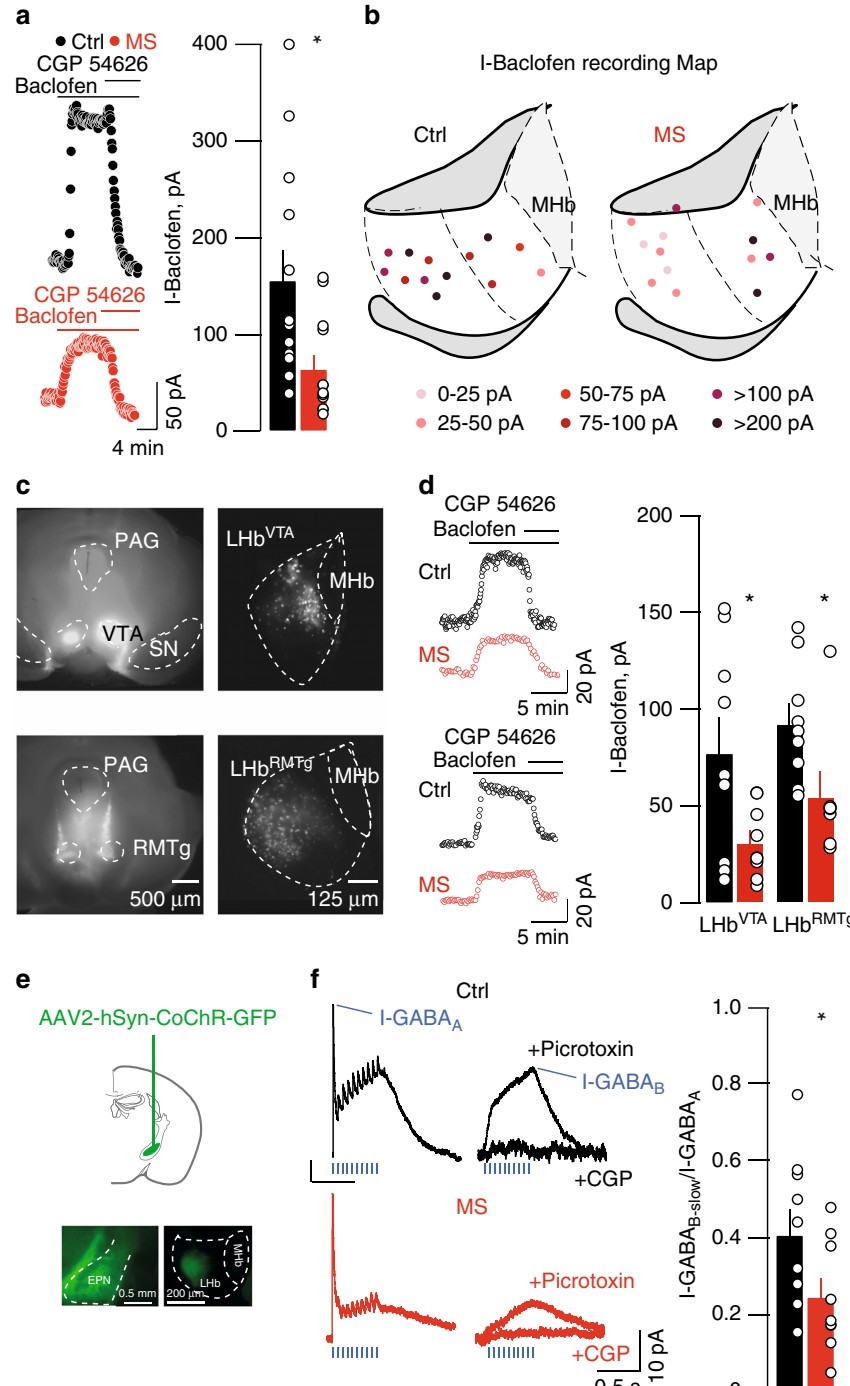

**Fig. 2** MS drives GABA$_B$-GIRK plasticity in LHb. **a** Sample traces, bar graph and scatter plots depicting CGP54626-sensitive I-Baclofen (I-Baclofen Ctrl vs. MS; $n_{mice}$ = 5/6; $n_{cells}$ = 13; unpaired t-test, $t_{24}$ = 2.67 *$p$ < 0.05). I-Baclofen was measured at steady state. **b** Territorial distribution of I-Baclofen showing MS-dependent reduction of GABA$_B$-GIRK signaling throughout the LHb. **c** Examples of the injection site of red retrobeads infused in the VTA (top) and in the RMTg (bottom). Right, retrogradely labeled LHb neurons projecting to VTA or RMTg. **d** Sample traces, bar graph and scatter plots depicting CGP54626-sensitive I-Baclofen in LHb$^{VTA}$ and LHb$^{RMTg}$ neurons (LHb$^{VTA}$: I-Baclofen Ctrl vs. MS; $n_{mice}$ = 2; $n_{cells}$ = 9; unpaired $t$-test, $t_{16}$ = 2.37 *$p$ < 0.05; LHb$^{RMTg}$: I-Baclofen Ctrl vs. MS; $n_{mice}$ = 2; $n_{cells}$ = 9 vs. 7; unpaired t-test, $t_{14}$ = 2.35 *$p$ < 0.05). **e** Stereotactic infusion of AAV2-hSyn-CoChR-GFP within the EPN, and opsin expression in terminals within the LHb **f** Opto-GABA$_A$-IPSCs and opto-GABA$_B$-IPSC sample traces and summary plot (Ctrl vs. MS; $n_{mice}$ = 6/3; $n_{cells}$ = 9 per group; unpaired $t$-test, $t_{16}$ = 2.21; *$p$ < 0.05). Values for opto-GABA$_A$-IPSC and opto-GABA$_B$-IPSC were taken at the maximal peak as indicated. Scale bars=4 min and 50 pA **a**, 500 and 125 µm **c**, 5 min and 20 pA **d**, 0.5 mm and 200 µm **e**, 0.5 s and 10 pA

(I-Baclofen) readily reversed by the specific antagonist CGP-54626 (Fig. 2a). I-Baclofen was significantly reduced throughout the LHb of MS mice (Fig. 2b, Supplementary Fig. 2a, b). The reduction of I-Baclofen, with no evident territorial specificity, suggests that MS reduces GABA$_B$-GIRKs independently of the

structures to which LHb neurons send their axons. To test this prediction, we used fluorescent tract tracing to retrogradely label LHb neurons projecting either to the ventral tegmental area (LHb$^{VTA}$) or the rostromedial tegmental nucleus (LHb$^{RMTg}$). These are prominent habenular-midbrain projections that, if

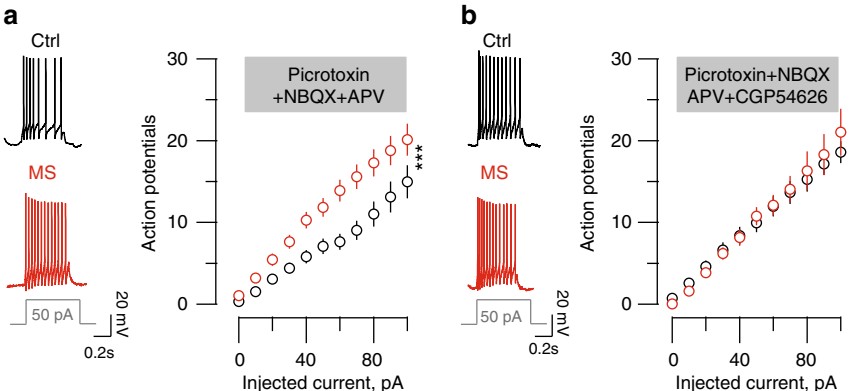

**Fig. 3** MS-induced LHb neurons hyperexcitability requires reduced GABA$_B$-GIRK signaling. **a** Left, sample traces for recordings in Ctrl and MS mice of a current-evoked firing (for a 50 pA step) in the presence of picrotoxin/NBQX. Graph representing the action potentials vs. injected current in all experimental groups. (picrotoxin/NBQX, $n_{mice}$ = 4/6; $n_{cells}$ = 20 per group; two-way ANOVA-RM, interaction $F_{(10;380)}$ = 3.52; ***$p$ < 0.001) **b** Same than **a** but in the presence of picrotoxin/NBQX/CGP54626, $n_{mice}$ = 5/6; $n_{cells}$ = 20 per group; two-way ANOVA-RM, interaction $F_{(10;380)}$ = 0.32). Scale bars=0.2 s and 20 mV

disrupted, contribute to the establishment of depressive-like symptoms[8, 12]. MS significantly reduced I-Baclofen in both LHb$^{VTA}$ and LHb$^{RMTg}$ neurons supporting a scenario in which GABA$_B$-GIRK signaling reduction occurs throughout the LHb, likely affecting multiple downstream targets (Fig. 2c, d).

GABA$_B$-R activation within the LHb leads to cell hyperpolarization through GIRK channels opening[11]. Along with the reduced I-Baclofen, MS also diminished GIRK currents. Indeed, responses generated by the intracellular dialysis of the G-protein activator GTPγS (I–GTPγS) and the bath application of the specific GIRK1/2 activator ML-297 were smaller throughout the LHb of MS mice (Supplementary Fig. 2c, d)[17, 18]. These results indicate that early-life stress, similarly to foot-shocks[11, 19], reduces GABA$_B$-GIRK function within the LHb.

Activation of the GABA$_B$-GIRK signaling requires GABA spillover from the presynaptic terminal[20]. A major input source of GABA onto LHb neurons arises from the entopeduncular nucleus (EPN). Furthermore, the reduction of EPN-originating inhibitory transmission within the LHb contributes to depressive-like symptoms[15, 21, 22]. We therefore set out to provide proof of principle that MS reduces, not only agonist-evoked GABA$_B$-R responses, but more precisely synaptically-relevant GABA$_B$-R function. To probe synaptically-activated GABA$_B$Rs, we transduced EPN neurons with a rAAV2-hSyn-CoChR-eGFP allowing for the expression of the excitatory opsin from *Chloromonas oogama* (CoChR; Fig. 2e)[23]. CoChR-expressing EPN terminals received trains of light-pulses (20 Hz), which elicited fast-GABA$_A$R and slow-GABA$_B$-R outward currents (I-GABA$_A$ and I-GABA$_{B-slow}$ respectively), the latter likely mediated by GABA spillover diffusing to perisynaptic GABA$_B$-Rs (Fig. 2f). We then computed the I-GABA$_{B-slow}$/I-GABA$_A$ ratios as a measure of the postsynaptic strength of inhibitory transmission. MS diminished the I-GABA$_{B-slow}$/I-GABA$_A$ ratios without significant alterations in miniature EPSCs or IPSCs, nor on presynaptic GABA$_B$-R function (Fig. 2f, Supplementary Fig. 3a–d). Altogether, MS diminishes postsynaptic GABA$_B$-GIRK signaling in LHb.

Thus, we predicted that MS-dependent LHb hyperexcitability results from the above-described GABA$_B$-GIRK reduction. Blocking GABA$_A$Rs, AMPARs and NMDARs left MS hyperexcitability unaffected. However, MS occluded GABA$_B$Rs antagonism-driven increased neuronal excitability observed in control slices (Fig. 1e, Fig. 3a, b; (Ctrl, aCSF/CGP54626, $F_{(10;410)}$ = 2.3 $p$ < 0.05; MS, aCSF/CGP54626, $F_{(10;410)}$ = 0.37 $p$ > 0.05; two-way ANOVA RM, interaction). This suggests that MS leads to LHb hyperexcitability via GABA$_B$-GIRK plasticity.

**Limiting habenular activity ameliorates depressive-like symptoms**. Is LHb hyperexcitability necessary for MS depressive-like phenotypes? MS-driven adaptations occur throughout the LHb (Fig. 2b–d). Therefore, we sought to limit hyperactivity of a large LHb neuronal population. Thus, we virally expressed the inhibitory designer receptors exclusively-activated by designer drugs (rAAV8-hSyn-HA-hM4Di-mCherry; Gi-DREADD) in the LHb (Fig. 4a, Supplementary Fig. 4a). Ex-vivo, bath application of the specific DREADDi ligand clozapine-N-oxide (CNO) generated a K$^+$-dependent outward current (I-CNO) and abolished neuronal firing (Supplementary Fig. 4b, c). MS reduced I-CNO, supporting a diminished GIRK function[24]. We next examined baseline neuronal activity using single unit recordings in anesthetized mice. The baseline firing rate of MS mice was higher than Control animals, an observation in line with the increased excitability reported in acute slices. The systemic injection of CNO diminished LHb neuronal firing in Gi-DREADD-expressing mice, and reduced the activity of MS-LHb neurons to values not statistically different from Control animals (Supplementary Fig. 4d, e). This validates this strategy to limit hyperexcitability in behaving mice. When exposed to escapable foot-shocks (shuttle box), in the absence of CNO MS mice (YFP/Gi-DREADD) expressed high failure rates (Fig. 4b). After a recovery period of 3 days, mice of all experimental groups received a single systemic injection of CNO ~15 min prior the test. CNO reduced the failure rate in the shuttle box only in Gi-DREADD-expressing MS mice. However, 5 days later, after complete clearance of CNO from the animal body, MS mice displayed the depressive state (Fig. 4b). Accordingly, in different sets of mice treated with CNO (see Methods), TST immobility and sucrose preference were comparable between Gi-DREADD-expressing MS mice and Controls (Fig. 4c, Supplementary Fig. 4f). In contrast, CNO treatment did not affect locomotor activity measured in the open field, arguing against compromised motor function (Supplementary Fig. 4f). Therefore, the chemogenetic reduction of LHb activity is sufficient to ameliorate MS-driven depressive-like symptoms.

Individuals presenting symptoms of depression following early-life stress poorly respond to antidepressants[25], heightening the need of alternative treatments. Deep brain stimulation (DBS) of the LHb ameliorates depressive symptoms, but its efficiency in the context of MS is unknown[12, 26]. When applied ex-vivo within the LHb, DBS-like stimulation (130 Hz)[12] reduced glutamate release, diminished neuronal firing, hyperpolarized neurons and decreased input resistance (Fig. 4d, e). Accordingly, in

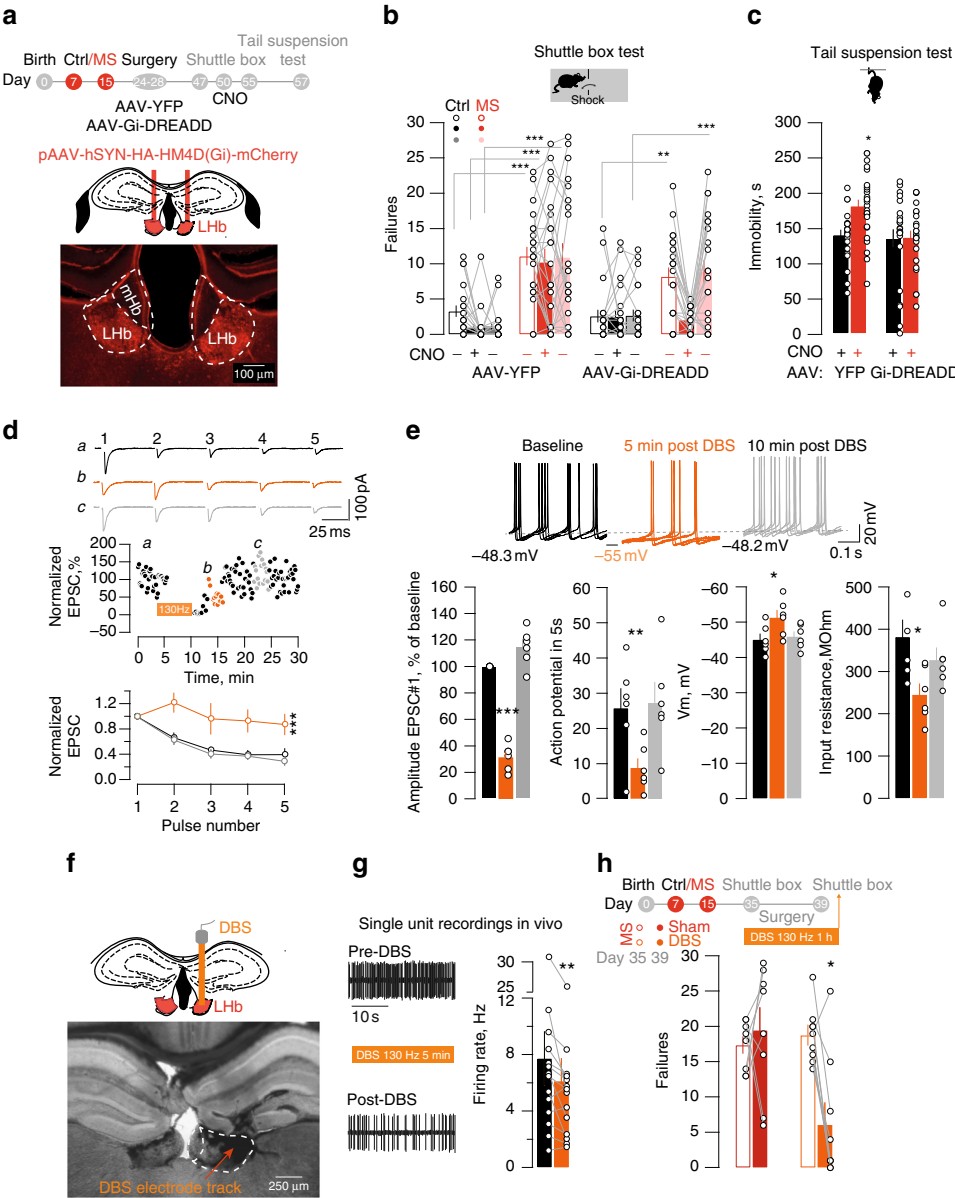

**Fig. 4** Chemogenetic and DBS approaches reduce LHb activity and ameliorate MS-dependent depressive-like symptoms. **a** Schematic and image for Gi-DREADD LHb expression. **b** CNO effects on failures in the shuttle box (AAV-YFP, Ctrl vs. MS and AAV-Gi-DREADD, Ctrl vs. MS; $n_{mice} = 21/22/23/26$; two-way ANOVA RM, interaction, $F_{(6,176)} = 3.29$, **$p < 0.01$). **c** Bar graph and scatter plot for TST immobility (AAV-YFP and AAV-Gi-DREADD, Ctrl vs. MS: $n_{mice} = 23/25/25/25$; two-way ANOVA, interaction, $F_{(1,94)} = 4$, *$p < 0.05$). **d** DBS effect on 1st EPSC and PPR (5 pulses, 20 Hz. Normalized eEPSC post-DBS, $n_{mice} = 2$; $n_{cells} = 6$; One-way ANOVA RM; $F_{(1.23,6.13)} = 99.65$, ***$p < 0.001$; Normalized EPSCs, $n_{mice} = 2$; $n_{cells} = 6$; Two-way ANOVA RM; DBS effect, $F_{(2,50)} = 25.1$, ***$p < 0.001$). **e** Sample I-clamp recordings (5 superimposed-sweeps), and DBS effects on action potentials, resting membrane potential and input resistance (Before vs. 5 min post-DBS vs. 10 min post-DBS, $n_{mice} = 2$; $n_{cells} = 6$: Action potential: $F_{(1.29,6.49)} = 13.5$, **$p < 0.01$; Vm: $F_{(1.15,5.77)} = 6.3$, *$p < 0.05$; Ri: $F_{(1.06,5.30)} = 8.3$, *$p < 0.05$; One-way ANOVA-RM) **f** DBS-electrode placement in LHb. **g** DBS-induced (130 Hz, 150 μA) reduction of activity in vivo (Firing before vs. post-DBS, $n_{mice} = 4$; $n_{cells} = 14$; paired t-test, $t_6 = 3.1$; **$p < 0.01$) **h** DBS-driven reduction of failures in the shuttle on MS mice (Sham vs. DBS; $n_{mice} = 8/9$; Two-way ANOVA RM, interaction, $F_{(1,15)} = 9.476$, **$p < 0.01$). Scale bars=, 100 μm **a**, 25 ms and 100 pA **d**, 0.1 s and 20 mV **e**, 250 μm **f**, 10 s **g**

anesthetized mice, DBS lowered LHb activity (Fig. 2f, g and Supplementary Fig. 4g). Therefore, DBS reduces neuronal activity by engaging both presynaptic and postsynaptic mechanisms. We then unilaterally implanted DBS electrodes in LHb (Fig. 4f), and examined its efficacy on MS-induced depressive-like behavior. DBS reduced the number of failures in the shuttle box (Fig. 4h). This highlights DBS as a strategy, complementary to pharmacology, to compensate for MS-driven LHb hyperexcitability and ameliorate depressive states.

## Discussion

Here we demonstrate that MS promotes LHb neuronal hyperactivity by reducing GABA$_B$-GIRK function. This hyperexcitability ultimately triggers depressive-like symptoms.

The GABA$_B$-GIRK signaling in the LHb tightly controls neuronal activity[11] and GABA$_B$-R internalization often occurs along with GIRK trafficking[11, 27, 28]. Consistently, we report that MS-driven reduced GABA$_B$-Rs function occurs along with diminished GIRK channels either expression or function. Internalization of

GABA$_B$-Rs and GIRKs requires the activity of phosphatases including PP2A, and a PP2A inhibitor presents antidepressant properties[11, 27, 28]. However, whether PP2A contributes to MS-driven GABA$_B$-GIRK plasticity in the LHb remains to be established.

In line with a reduction of I-Baclofen, we observed a diminished I-GABA$_{B-slow}$/I-GABA$_A$ ratios at EPN-to-LHb synapses. These findings provide new insights on (i) Physiological patterns of presynaptic activity to elicit synaptic GABA$_B$-Rs in the LHb; (ii) MS-mediated plasticity of synaptically-evoked GABA$_B$-Rs. However MS-driven GABA$_B$-Rs reduction may also occur at synaptic inputs other than the EPN.

Our data indicate that MS does not significantly affect mIPSCs, however these events stem from unidentified inputs, leaving open the possibility that MS also engages input-specific changes of fast inhibitory transmission[15, 21]. In addition, the learned helplessness depressive state potentiated fast excitatory transmission onto LHb$^{VTA}$ neurons[12]. We cannot completely rule out that a similar adaptation also occur in MS mice. The detection threshold for MS plasticity of mEPSCs might be lowered by the different animal model of depression employed or, alternatively, by circuit-specific adaptations yet to be identified. Overall, plasticity of GABA$_B$-GIRKs in the LHb emerges as a general cellular substrate underlying stressor-driven depressive phenotypes as it is common to acute traumatic events and chronic stressful conditions such as MS[11, 19].

Reduced GABA$_B$-GIRK signaling mediates LHb neuronal hyperactivity[11]. Considering that MS-driven plasticity occurs in VTA and RMTg-projecting LHb neurons, MS cellular adaptations may remodel target midbrain circuits highlighting LHb-to-midbrain relevance in mood disorders[3, 8, 12]. The habenular output connectivity is however complex indicating that LHb-receiving structures other than the midbrain may contribute to the MS-driven behavioral phenotypes. This is indeed likely, as the inactivation of the LHb fails to rescue aberrant dopamine neurons activity in the chronic mild stress model of depression[29]. Altogether, stress-driven depressive-like symptoms emerge from the dysfunction of complex and parallel neuronal circuits (i.e., LHb-related and midbrain-related)[11, 29–31].

MS-dependent hyperactivity can be limited by chemogenetic and DBS approaches, which in turn are efficient to ameliorate MS-mediated depressive-like phenotypes.

Our data provide an extensive description of Gi-DREADD efficiency at the single-cell level ex-vivo and in-vivo within the LHb. We report that Gi-DREADD activation in vivo in the LHb reduces, but does not silence, neuronal activity. This highlights Gi-DREADD efficacy in rescuing aberrant activity, without abruptly altering the physiology of neural systems. These properties may differ according to neuronal populations, viral vector properties or promoter employed.

In addition to a chemogenetic-based strategy we also investigated the use of DBS approaches to reduce MS hyperactivity in the LHb. The high-frequency DBS protocol reported in our work was previously shown to also ameliorate depressive symptoms emerging in congenital learned helplessness rats[12]. Our data suggest that DBS efficiency not only relies on reduced presynaptic release[12], but also on postsynaptic mechanisms. The DBS-mediated modulation of LHb activity is however transient, suggesting that the MS depressive-like phenotypes can re-emerge after the intervention[12]. This indicates that DBS efficacy stems from its pre/postsynaptic-mediated reduction in LHb neuronal activity rather than on the MS-driven plasticity expression mechanisms (i.e., GABA$_B$-GIRK plasticity).

Altogether, this study provides mechanistic insights underlying MS-induced adaptations in the LHb. Moreover, limiting LHb neuronal activity has potential therapeutic relevance in alleviating affective symptoms of neuropsychiatric disorders.

## Methods

**Experimental subjects and MS paradigm**. All procedures were used in accordance with the guidelines of the French Agriculture and Forestry Ministry for handling animals (Committee Charles Darwin #5, University Pierre et Marie Curie, Pairs). Part of the current study was carried out in the Department of Fundamental Neuroscience (Lausanne, Switzerland) under license and according to regulations of the Cantonal Veterinary Offices of Vaud and Zurich (Switzerland). Pregnant dams C57Bl/6J were received at the gestational stage E13–18 (Janvier Laboratories, France). Mothers were housed 2 per cages with access of food and water ad libitum. After birth, pups of either sex remained untouched until postnatal day (P) 7. At P7, litters were randomly divided in 2 groups. The maternal separation group consisted of pups removed from their litter and isolated in small compartments for 6 h per day (light phase 0800:1900 hours) repeated from P7 to P15 and followed by an early weaning at P17. During the separation, animals were maintained in heating plate and water was provided, maintaining constant temperature and humidity. The control group consisted of mice from independent litters, which were not manipulated until the regular weaning at P21 except during cage changing. During cage changing some old bedding and nest were transferred into the new cage in order to limit novelty stress. After weaning, mice where separated by sex and housed 6 per cage. Experiments were performed in mice aged 4–8 weeks.

**Surgery**. Animals, aged at least 24 days were anesthetized with Ketamine (150 mg kg$^{-1}$)/Xylazine (100 mg kg$^{-1}$ i.p.) before bilateral injection of rAAV8-Hsyn-Gi-DREADD-mCherry (University of Pennsylvania, US) in the LHb at the following coordinates (from bregma, in mm): A-P: –1.45; M-L:±0.45; D-V: –3.1. After three weeks, mice were subjected to CNO i.p injection (1 mg kg$^{-1}$) for the DREADD activation. For optogenetic experiments rAAV2.1-hSyn-CoChr-eGFP (University of North Carolina, US) was infused in the entopeduncular nucleus (from bregma, in mm: A-P: –1.25; M-L:±1.8; D-V: –4.65). Recordings were performed 3 weeks after surgery. The injection sites were carefully examined and only animals with correct injections were kept for behavioral and electrophysiological analysis. DBS electrodes were unilaterally implanted using similar procedures and coordinates in the LHb. DBS electrodes were chronically implanted using a Superbond resin cements (Sun medical, Japan). For the experiment analyzing the output specificity of I-Baclofen, mice were bilaterally injected with a mixture of herpes simplex virus (McGovern Institute, USA) expressing enhanced GFP and red retrobeads (Lumafluor, US) into the RMTg or the VTA. The following coordinates were used (RMTg: from bregma, in mm: A-P: –2.9; M-L:±0.5; D-V: –4.3; VTA: from bregma, in mm: A-P: –2.4; M-L:±0.65; D-V: –4.9). Recordings from fluorescent LHb neurons were performed ± 12 days following the surgery, and injection site were verified using the retrobeads labeling.

**Behavioral paradigm**. All experimental behaviors were performed during the light phase and experimenters were blind of their experimental group.

The shuttle box test was performed in a shuttle box (13 × 18 × 30 cm$^3$) equipped with an electrified grid floor and a door separating the two compartments. The test session consisted of 30 trials of escapable foot-shocks (10 s at 0.1–0.3 mA) separated by an interval of 30 s. The shock terminated any time that the animal shuttled in the other compartment. Failure is defined as the absence of shuttling to the other compartment within the 10 s shock delivery.

The tail suspension test was performed with mice being suspended by their tails with adhesive tape for a single session of 6 min. Immobility time of each animal was scored online by the experimenter. Mice were considered immobile only when they suspended passively and motionless.

The sucrose test preference was performed with mice being single-housed and habituated with two bottles of 1% sucrose for 2 days. At day 3 (test day) mice were exposed to two bottles filled with either 1% sucrose or water for 24 h. The sucrose preference was defined as the ratio of the consumption of sucrose solution vs. total intake (sucrose+water) during the test day and expressed as a percent.

Behavioral experiments in DREADD-injected animals were performed three weeks after viral infusion. For the shuttle Box, the tail suspension test, and the locomotor activity all the groups (YFP or DREADDi injected animals) were injected 15 min with CNO i.p. (1 mg kg$^{-1}$). For sucrose preference experiments, all groups were injected with CNO i.p. (1 mg kg$^{-1}$) every 3 h for the extent of the preference session (24 h) to maintain a constant DREADD-mediated inhibition.

**Electrophysiology**. For in vitro recordings, animals were anesthetized with ketamine and xylazine (i.p. 150 mg kg$^{-1}$ and 100 mg kg$^{-1}$, respectively). Coronal brain slices (250 μm) containing the LHb were prepared in bubble ice-cold 95% O$_2$/5% CO$_2$-equilibrated solution containing: 110 mM choline chloride; 25 mM glucose; 25 mM NaHCO$_3$; 7 mM MgCl$_2$; 11.6 mM ascorbic acid; 3.1 mM sodium pyruvate; 2,5 mM KCl; 1.25 mM NaH$_2$PO$_4$; 0.5 mM CaCl2. Slices were then stored at room temperature in 95% O$_2$/5% CO$_2$-equilibrated artificial cerebrospinal fluid (ACSF) containing: 124 mM NaCl; 26.2 mM NaHCO3; 11 mM glucose; 2.5 mM KCl; 2.5 mM CaCl$_2$; 1.3 mM MgCl$_2$; 1 mM NaH2PO4. Recordings (flow rate of 2.5 ml min$^{-1}$) were made under an Olympus-BX51 microscope (Olympus, France) at 30 °C. Currents were amplified, filtered at 5 kHz and digitized at 20 kHz. Access resistance and input resistance were monitored by a step of –4 mV (0.1 Hz). Experiments were discarded if the access resistance increased >20%.

The internal solution used to examine GABA$_B$ and/or GIRK currents and neuronal excitability contained: 140 mM potassium gluconate, 4 mM NaCl, 2 mM MgCl$_2$, 1.1 mM EGTA, 5 mM HEPES, 2 mM Na2ATP, 5 mM sodium creatine phosphate, and 0.6 mM Na3GTP (pH 7.3 with KOH). The liquid junction potential was ~12 mV. When we measured the synaptic inhibitory or excitatory release, the internal solution contained: 130 mM CsCl; 4 mM NaCl; 2 mM MgCl$_2$; 1.1 mM EGTA; 5 mM HEPES; 2 mM Na2ATP; 5 mM sodium creatine phosphate; 0.6 mM Na3GTP; and 0.1 mM spermine. The liquid junction potential was −3 mV. Whole-cell voltage-clamp recordings were achieved to measure GABA$_B$-GIRK currents in aCSF only. For agonist-induced currents, changes in holding currents in response to bath application of baclofen (100 μM) were measured (at −50 mV every 5–10 s). The plotted values correspond to the difference between the baseline and the plateau (for the baclofen and ML297 experiments) or the difference between the plateau and the value of holding current after barium (for the I-GTP-γS) GABAB-GIRK currents were confirmed by antagonism with 10 μM of CGP54626. When stated, 100 μM of GTP-γS was added to the internal solution in place of Na3GTP. Plateau currents were then reversed by 1 mM Barium application, a selective inhibitor of K+channels. Changes in holding currents in response to GIRK agonist were measured (at −50 mV every 5–10 s) by bath application of ML-297 (50 μM), a Selective GIRK1/2 channel activator then reversed by 1 mM Barium application. Synaptic GABA$_B$ slow IPSCs were optically evoked by trains of 10 pulses delivered at 20 Hz through a 470 nm LED. The fast GABA amplitude correspond to the amplitude of the first pic of the train, the slow GABA current instead were measured after picrotoxin bath application, and correspond to the I-max. Miniature excitatory postsynaptic currents (mEPSCs) were recorded in voltage-clamp mode at −60 mV in the presence of bicuculline (10 μM), AP5 (50 μM) and tetrodotoxin (TTX, 1 μM). Miniature inhibitory postsynaptic currents (mIPSCs) were recorded (−60 mV) in the presence of NBQX (20 μM) AP5 (50 μM) and TTX (1 μM). EPSCs were evoked through an ACSF-filled monopolar glass electrode placed in the LHb. For the experiments in which high-frequency stimulation trains were used to determine presynaptic release probability (5 pulses at 20 Hz), QX314 (5 mM) was included in the internal solution to prevent the generation of sodium spikes.

Current-clamp experiments were performed using a series of current steps (from −80 to 100 pA or when the cell reached a depolarization block) injected to induce action potentials (10-pA injection current per step, duration of 500 ms). Cells were maintained at −55 mV throughout the experiment. When testing changes in tonic firing, cells were depolarized to obtain stable firing activity in current-clamp mode.

When in vivo single unit recordings were performed, mice were anesthetized with isoflurane (induction: 2%; maintenance: 1–1.5%) using an anesthesia device for small animals (Univentor 410, Malta). We placed the mice in the stereotaxic apparatus (Kopf, Germany) and their body temperature was maintained at 36 ± 1 °C using a feedback-controlled heating pad (CMA 450 Temperature Controller, USA). The scalp was retracted and one burr hole was drilled above the LHb (A-P: −1.3/−1.6; M-L:±0.4/0.5) for the placement of a recording electrode. Single unit activity of neurons located in the LHb (Ventral 2.3–3.2 mm to cortical surface) was recorded extracellularly by glass micropipettes filled with 2% pontamine sky blue dissolved in 0.5 M sodium acetate (impedance 3–6 MΩ). Signal was pre-amplified (DAM80, WPI, Germany), filtered (band-pass 500–5000 Hz) amplified (Neurolog System, Digitimer, UK), displayed on a digital storage oscilloscope (OX 530, Metrix, USA), and digitally recorded. Experiments were sampled on- and off-line by a computer connected to CED Power 1401 laboratory interface (Cambridge Electronic Design, Cambridge, UK) running the Spike2 software (Cambridge Electronic Design). Single units were isolated and identified according to previously described electrophysiological characteristics (Meye et al.[8]) including a broad triphasic extracellular spike (>3 ms), and a tonic regular, tonic irregular or bursting spontaneous activity.

Isolated LHb neurons were recorded for 5 min to establish the basal spontaneous firing rate. CNO was administered i.p. (1 mg kg$^{-1}$) and the firing activity of the neuron was monitoring every 5 m for total 40 m. When CNO was administered, only one cell was recorded per mouse.

DBS experiments were performed with a modified double barrel system allowing to stimulate in close proximity of the recording site: the stimulating electrode was attached to the recording one by using glass barrels (the 2 electrodes form an angle of ±30°). The stimulating tip was glued above the recording tip (<300 μm). A stable spontaneous firing rate was recorded for 5 min before to start the DBS protocol (total duration: 2–5 m; Train pulses: 7; ITI: 40 ms; Frequency: 130 Hz; Intensity: 150 μA). The firing activity recorded immediately after the protocol was compared with the respective baseline.

At the end of each experiment, the electrode placement was marked with an iontophoretic deposit of pontamine sky blue dye (−80 μA, continuous current for 35 min). Brains were then rapidly removed and fixed in 4% paraformaldehyde solution. The position of the electrodes was microscopically identified on serial sections (60 μm).

**Deep brain stimulation**. MS mice for DBS experiments were first preselected on the basis of their failure rate in the Shuttle box test (A cutoff of 12 failures was used for the preselection). In total 50 mice were tested, and 17 of these animals met the criteria. Standard surgical procedures were used to implant bipolar concentric electrodes unilaterally into the LHb (coordinates −1.45 mm AP, ±0.45 mm ML and −3.1 mm DV). After 5 days recovery from surgery, DBS or no (Sham) stimulation was applied for 1 h (seven stimulus trains of 130 Hz, separated by 40 ms intervals; 150 μA intensity) prior testing each mouse in the shuttle box test.

**Analysis and drugs**. All drugs were obtained from Abcam (Cambridge, UK) and Hello Bio, and Tocris (Bristol, UK) and dissolved in water, except for TTX (citric acid 1%), ML297 and CNO (DMSO). Online/offline analysis were performed using IGOR-6 (Wavemetrics, US) and Prism (Graphpad, USA). Data analysis for in vivo electrophysiology was performed off-line using Spike2 (CED, UK) software. Sample size required for the experiment was empirically tested by running pilots experiments in the laboratory. While behavioral experiments were run in a single-to-triple trial, electrophysiological experiments were replicated at least five times. Experiments were replicated in the laboratory at least twice. Animals were randomly assigned to experimental groups. Data distribution was assumed to be normal, and single data points are always plotted. Compiled data are expressed as mean ± s.e.m. All groups were tested with Grubbs exclusion test (limit set at 0.05) to determine outliers. Significance was set at $p < 0.05$ using Student's t-test two-sided, Kolmogorov–Smirnov test, one-way, two-way or three-way Anova with multiple comparison when applicable.

**Data availability**. All relevant data are available from the authors upon request.

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

### Acknowledgements

This work was supported by the European Research Council (Starting grant SalienSy 335333) to M.M. We are grateful to M. Creed and C. Lüscher for support on DBS approach. We thank C. Bellone, M. Creed, E. Schwartz, C. Lüscher. and the Mameli Laboratory for feedback on the manuscript. M.M. is a member of the Fens-Kavli Network of Excellence.

### Author contributions

A.T. performed and analyzed in vitro recordings together with K.V. and M.M. S.L. performed in vivo recordings. A.T. performed behavioral experiments with M.M. and S.L. M.M. and A.T. designed the study and wrote the manuscript.

### Additional information

**Competing interests:** The authors declare no competing financial interests.

