## [Peer Review File · Nature Communications]

Reviewers' expertise:

Reviewer #1: Systems neuroscience approaches, circuits related to depression, mouse models;

Reviewer #2: DBS, psychiatric disorders;

Reviewer #3: Systems neuroscience approaches, synaptic plasticity in depression, mouse models.

Reviewers' comments:

Reviewer #1 (Remarks to the Author):

In this paper, the authors examine how neural activity in the lateral habenula (LHb) changes following the induction of a depression-like state with maternal deprivation (MD). They report that MD induces GABAB-GIRK-dependent hyperexcitability in LHb, and that chemogenetic or DBS reduction of this hyperactivity results in a reduction of depression-like behavior. These results are likely to be of broad interest.

Major points

1. The authors show a significant effect of MD on two locomotor assays (shuttle box and TST) and one hedonic assay (sucrose preference) of depressive behavior, and do not detect a significant effect of MD on a locomotor control assay (open field). However, while they report a significant effect of inhibitory DREADDS and DBS on behavior in the shuttle box and TST, they do not report the impact of either of these manipulations on open field or sucrose preference behavior. This leaves open the possibility that the effect of this manipulation is simply a non-specific increase in locomotor activity, not a reduction in depressive state. The open field data would be sufficient to address this question, but I recommend that the authors conduct sucrose preference as well. Even if sucrose preference is not significant, this data would be of great interest – a negative result could point towards separable pathways for effortful behavior in an aversive situation and reward sensitivity.
2. These results need to be integrated (in the discussion) with prior work by Malinow implicating enhanced excitatory transmission and reduced GABAA-mediated inhibitory transmission at the entopeduncular (EP)-lateral habenula synapse in depression, which were not detected in the current study. What are the possibilities for explaining these divergent results? Depression model, type of LHb neuron under study, type of EP neuron infected, stimulation protocol?
3. Non-specific AAV vector spreads widely beyond the habenula if extreme care is not taken with validation of targeting. Did you only include animals where expression was strictly confined to the habenula? The PVT is adjacent to this structure, and it is also an important brain region for appetitive and aversive motivated behaviors; there does appear to be some expression in this region (Fig 1a).

Minor points

1. The T statistic for the sucrose preference test (page 2, reporting checklist; Fig 1d) does not match the p value. One of these two figures must be incorrect, and this affects whether sucrose preference was significantly different between the MD and control groups.
2. Should cite Shabel et al., 2014

Reviewer #2 (Remarks to the Author):

In this manuscript, the authors examine the well-characterized habenula hyperactivity model of depression in a mouse model of early life stress. The experiments are very straightforward, with mice exposed to MD showing habenula hyperexcitability due to GABA B GIRK reduction. However, the interpretation should be tempered. i.e., although they showed that the habenula is hyperexcitable in the MD mice, and that shutting off the habenula reverses the impact of MD, it is not clear if this is an offsetting effect (e.g., studies showing that with CMS there is overdrive of the iIPFC with no impact on the habenula; Moreines et al.). Since the state of the habenula in the normal mouse is not one of complete inactivity, the DREADD experiments do not show a selective reversal of the deficit state. It would also be important to examine DBS effects on the GABA B GIRK system.

Reviewer #3 (Remarks to the Author):

Inescapable stress has garnered much interest as a translational paradigm to study depression in animal models. Building upon the results of their recent Nature Medicine publication, Tehenio et al. find that maternal deprivation (a form of inescapable stress) may result in a depressive-like phenotype through downregulation of GABABR-GIRK conductances, and hyperactivity of the lateral habenula. Furthermore, the group goes on to nicely demonstrate that decreasing habenular hyperexcitability can lead to improvement in measures of anhedonia.

While a potentially interesting study, my main concern is that the authors have not taken advantage of the article format, using only 1000 words out of their 5000 limit. It is therefore difficult for me to review this manuscript with full consideration, as there is little to none of introduction, justification of experimental design, explanation of results, or discussion. Consequently, I have many unanswered questions that would likely be resolved with more substantive writing. If the group utilized the article format to its fullest extent, and took care to include some conspicuously absent controls then I believe that this manuscript would be suitable for publication in Nature Communications, and of interest to a wide audience of systems neuroscientists, and circuit psychiatrists.

The first half of the paper extends results from their recently published study, in which the group demonstrates that various forms of inescapable stress result in hyperexcitability of the LHb and a depression-like phenotype. As such it is not surprising that another stressor, maternal deprivation, may lead to a depressive-like phenotype in a similar manner. My main concern with these experiments is the reliance solely on pharmacology, and forgoing the immunological and genetic techniques utilized in their previous work. In particular, the

agent used to implicate GIRK receptors, GTPyS, is a pleiotropic agonist that will have an effect on a variety of different GPCRs.

I am also concerned that the behavioral testing was not performed blind to genotype, especially the TST which relies on subjective observer rating. Can the authors comment on why the behavioral analysis and testing was not done blindly?

It is interesting that the MD-dependent reduction of GABAB-GIRK signaling is observed throughout the LHB. Can the authors comment on why MD appears to have such a wide-spread effect in the LHB? Based on the current literature the negative valence of a hyperexcitable habenula is mediated by its outputs to the RMTg/VTA. It would be helpful to demonstrate that the neurons affected by MD do indeed project to these nuclei.

Other points:

1. Insufficient detail is given in the methods for the MD protocol considering the entirety of the paper relies on it. According to reference #2 there are many different established protocols, with differing results. From the methods it seems that this group is not doing a MD protocol, but rather a BMS protocol, can the authors clarify?
2. The authors state that the control groups were untouched. Does this mean that cages were not changed? No bedding changes? Since the authors use ref 2 as a justification of their MBS protocol it would be helpful to use the same terminology and designations as in the cited review.
3. Ketamine was used as anesthesia, however ketamine administration has antidepressant effects that may cloud the authors results. Can the authors comment on this?
4. The authors should include the series and input resistances of the cells they record from. These measurements are crucial to determine the quality of the recordings, especially when measuring absolute current amplitudes across multiple cells/animals.
5. Additionally, please include the resting membrane potentials for the current clamp experiments. If LHB neurons are hyperexcitable through loss of a K⁺ conductance, this should be reflected in a difference between the resting membrane potentials in LHB neurons of Ctrl and MD animals.
6. For fig 1f and 1g it is not clear when during the recordings the current amplitudes are measured. For fig 1f I assume it's when I-baclofen reaches steady state. For 1g please indicate on the traces where I-Gaba fast/slow are being measured.
7. For fig1e the figure legend states a 50pA step was used, but it looks like several different amplitude steps were used?

The second half of the paper demonstrates that reducing habenula activity leads to improvement in the depression-like phenotype. These results are intriguing, especially in light of future treatments for affective disorders. However, again this manuscript is difficult to understand as the authors have not written in enough detail to sufficiently justify their experimental design. It is difficult for me to comprehend the full methodology of the experiments from only the cartoon schematics included in the figures. For example, it is unclear to me for how when and how long CNO was applied during behavioral testing. Furthermore, the behavioral analyses were not done blind to experimental conditions.

Other points:

1. Why is there a reduction in failures with CNO application in the Ctrl AAV-YFP group, and

why does this reduction persist even after discontinuation of CNO. If this is due to learning how to escape the shock, why is this effect not seen in the Ctrl DREADD group? Why is there no effect of CNO on failure rate in the Ctrl DREADD condition?

2. Supp fig3d demonstrates CNO application leads to a reduction in firing rate in the Ctrl DREADD condition. Why is this reduction in firing rate not correlated with a reduction in failures in the shuttle box experiments?

3. Why is the CNO mediated current lessened in the MD animals? Is this explained by differences in resting membrane potential?

4. In sup Fig 3c if those recordings are done in control mice, a complementary experiment with MD mice should be shown. The efficacy of CNO to reduce the absolute firing rate should be assessed in MD animals.

5. Supp fig 3d shows the relative firing rates, can the authors include the absolute firing rates? Even though the relative reduction appears to be similar, I would like to see that CNO application in the MD animals leads to an absolute firing rate comparable to baseline in the Ctrl. If the firing rate in the MD group after CNO application is still on average greater than in the Ctrl group would you still then expect increased failures in the shuttle box?

6. If I-CNO is decreased in MD animals compared to Ctrl, why do the authors observe the same time course and relative decrease in habenular firing rate in both experimental groups after CNO application?

7. The cartoon schematic and histology in Fig 2f indicates the the DBS electrodes were implanted unilaterally, however the methods mention bilateral placement of DBS electrodes. Can the authors clarify?

8. The use of DBS in the lateral habenula to reduce symptoms of depression has been successfully attempted before in both animals and human patients. Can the authors compare their DBS protocol with those previously used, and place their manuscript in context with the previous literature?

9. The author's DBS protocol appears to reduce excitatory signaling through a presynaptic mechanism, leading to a reduction in release probability. Can the authors please comment more on how they believe DBS is leading to decreased habenular activity. What is the cause of the observed decrease input resistance?

10. The authors use a 5min DBS (or stimulus train) protocol which produces a transient depression, to initially characterize the effects of DBS. However, during the in vivo experiment they appear to switch to a longer one hour protocol. Why wasn't the one hour protocol used in the initial characterization? I believe this is important as the effects on neuronal activity of 5 min vs 1 hour of DBS could be different.

Point by point response to Reviewers.

Reviewer #1

1. while they report a significant effect of inhibitory DREADDS and DBS on behavior in the shuttle box and TST, they do not report the impact of either of these manipulations on open field or sucrose preference behavior.

We thank the reviewer for raising this point. We performed new experiments to test whether Gi-DREADD activation in the LHb affects locomotion and sucrose preference in the two experimental groups. We find that locomotion remains similar across experimental groups arguing against motor dysfunctions. On the other hand, we report that sucrose preference was similar between control and MS mice when activating Gi-DREADD consistent with the other depressive-like symptoms we reported. These new data are now included in Supplementary Figure 4.

2. These results need to be integrated (in the discussion) with prior work by Malinow implicating enhanced excitatory transmission and reduced GABAA-mediated inhibitory transmission at the entopeduncular (EP)-lateral habenula synapse in depression, which were not detected in the current study. What are the possibilities for explaining these divergent results? Depression model, type of LHb neuron under study, type of EP neuron infected, stimulation protocol?

We agree with the reviewer's comment. We now included in the manuscript the seminal work from the Malinow laboratory. Our data show that miniature EPSCs and IPSCs were not affected by maternal separation. Our result may diverge from the initial observation that congenital Learned Helplessness rats present higher glutamatergic and lower EPN-driven GABA transmission. The major difference between these studies is the animal model used for depression. In the Learned Helplessness model of animals are pre-selected accordingly to their failure rate in the shuttle box. In contrast, in our work MS mice were not preselected leading to interindividual variability in showing depressive phenotypes. Another aspect that needs to be considered is that the reported changes in fast neurotransmission are input or output specific. We may have missed those as our readout stems from many inputs terminals within the LHb. This may altogether wash the effect at the synaptic level, yet leaving the GABA_B reduction as a major readout.

3. Non-specific AAV vector spreads widely beyond the habenula if extreme care is not taken with validation of targeting. Did you only include animals where expression was strictly confined to the habenula? The PVT is adjacent to this structure, and it is also an important brain region for appetitive and aversive motivated behaviors; there does appear to be some expression in this region (Fig 1a).

We agree with the reviewer, as this is a very critical point for the interpretation of the results, especially at the behavioral level. As also stated in the material and methods, we included in the analysis only animals with localized expression in the LHb. This is depicted in Supplementary Figure 4.

Minor points

1. The T statistic for the sucrose preference test (page 2, reporting checklist; Fig 1d) does not match the p value

We apologize for this mistake. The T value was erroneously reported. The real numbers are $t_{30}=2.121$, $p=0.0423$, and this is now modified in the reporting checklist.

2. Should cite Shabel et al., 2014

Done

Reviewer #2

1. ...the interpretation should be tempered. i.e., although they showed that the habenula is hyperexcitable in the MD mice, and that shutting off the habenula reverses the impact of MD, it is not clear if this is an offsetting effect. Since the state of the habenula in the normal mouse is not one of complete inactivity, the DREADD experiments do not show a selective reversal of the deficit state.

The reviewer raises a very interesting point. If DREADD activation would totally silence neuronal activity in the LHb, the risk of an offsetting effect would be high. However, the baseline firing rate we recorded in vivo indicates a residual activity after Gi-DREADD activation. This is also shown in the new data set in Supplementary figure 4e where we show CNO driven reduction but not silencing of activity. This was the rationale of using DREADD technology rather than lesioning approaches, which would have raised indeed this type of issues.

2 It would also be important to examine DBS effects on the GABA B GIRK system.

The reviewer raises a point that the authors thought important already at the first submission. We did heavily attempt to record from mice previously implanted with DBS electrodes. However, due to the detachment of the implant during the slice preparation lead to a very low quality of the tissue. Neurons were extremely unhealthy. This considered, we ask the reviewer to keep this difficulty in mind, and that even if we could manage in performing such recordings, we would not be at ease about the solidity of the data obtained in such compromised tissue.

We do nevertheless discuss this issue in the manuscript, as GABA_B-GIRK may not be rescued after the DBS, which is likely to only acutely reduce firing rate, without acting on the expression mechanisms of MS-mediated plasticity.

Reviewer #3

1. While a potentially interesting study, my main concern is that the authors have not taken advantage of the article format, using only 1000 words out of their 5000 limit.

We acknowledge the reviewer's suggestion. The manuscript was transferred from another Nature Journal in a format appropriate for the previous submission, and that was the reason of his compactness. We made effort in this resubmission to enlarge the text (introduction and discussion mainly), and re-format the figures.

2. the agent used to implicate GIRK receptors, GTPγS, is a pleiotropic agonist that will have an effect on a variety of different GPCRs.

We agree with the reviewer on the broad actions of GTPγS. Our objective with this set of experiment was to test whether MS leads to plasticity of GABA_B-Rs only or of the entire GABA_B-GIRK machinery. We strengthened our results that adaptations also occur downstream GABA_B-Rs performing a new set of experiments by using the specific GIRK1/2 activator ML-297 (Kaufman et al., 2013; Nimitvilai et al., 2017) in slices from Control and MS mice. As reported in the new Supplementary Figure 2d, ML-297 responses were reduced in slices from MS mice compared to control slices, supporting plasticity of GABA_B and GIRK component.

3. I am also concerned that the behavioral testing was not performed blind to genotype, especially the TST which relies on subjective observer rating. Can the authors comment on why the behavioral analysis and testing was not done blindly?

We apologize for the confusion. The statement used in the reporting checklist was mainly referred to the electrophysiological recordings. Two operators, blind of the experimental groups, performed and analyzed behaviors.

4. It would be helpful to demonstrate that the neurons affected by MD do indeed project to these nuclei.

The reviewer raises a very interesting issue. To test whether LHb neurons affected by MS are projecting to specific output structures, we performed a new series of experiment, where we patched retrogradely-labeled neurons projecting to either RMTg or VTA. We show in the new Figure 2 that GABA-B-GIRK currents decreased in MS mice independently of these projection targets.

Other points:

1. From the methods it seems that this group is not doing a MD protocol, but rather a BMS protocol, can the authors clarify?

The literature around maternal deprivation, and maternal separation is vast, and many protocols different from one another have been used. The reviewer is nevertheless correct, and we used a *maternal separation* protocol by definition. According to the reviewer's suggestion we preferred to be consistent with the published literature and we decided to use MS (maternal separation) throughout the manuscript. We furthermore included a better description of the protocol in the text.

2. The authors state that the control groups were untouched. Does this mean that cages were not changed? No bedding changes?

We apologize for the confusion. The bedding was regularly changed. Control animals were not separated from their mothers however.

3. Ketamine was used as anesthesia, however ketamine administration has antidepressant effects that may cloud the authors results. Can the authors comment on this?

The reviewer raises a very good point, particularly for the data concerning behavioral analysis in mice that underwent surgery. However, we do report and detect a depressive phenotype in MS mice infused with control viruses. This indicates that at least in this experimental setting the ketamine exposure was not producing, overall, a complete anti-depressant effect. An interesting observation is that the antidepressant actions of R,S-ketamine at high concentrations (used in our study) seem to fade away in about 24 hours. Given that our experiments are performed weeks after surgery, ketamine antidepressant actions would be already gone by the time we made the behavioral assessment.

4. The authors should include the series and input resistances of the cells they record from.

These analysis is now included in the newly modified Supplementary Figure 2a and b.

5. Additionally, please include the resting membrane potentials for the current clamp experiments.

We thank the reviewer for raising this point. We included the V_{rest} from recorded neurons in the new Supplementary Figure 1d, but this remains comparable between groups. The absence of an effect on V_{rest} can be due to either a lack of impact on V_{rest} after reduction of GIRK signal (similar to Lecca et al., 2016). Alternatively, the

measurement of V_{rest} in whole-cell mode does not allow to precisely resolve small modifications in V_{rest} occurring after MS.

6. For fig 1f and 1g it is not clear when during the recordings the current amplitudes are measured. For fig 1f I assume it's when I-baclofen reaches steady state. For 1g please indicate on the traces where I-Gaba fast/slow are being measured.

Done. This is now better explained in the figure caption.

7. For fig1e the figure legend states a 50pA step was used, but it looks like several different amplitude steps were used?

The reviewer is correct. We modified the figure caption accordingly.

8. It is difficult for me to comprehend the full methodology of the experiments from only the cartoon schematics included in the figures. For example, it is unclear to me for how when and how long CNO was applied during behavioral testing. Furthermore, the behavioral analyses were not done blind to experimental conditions.

We apologize with the reviewer for the lack of clarity. We now revised the manuscript providing more experimental detail in the text. We also apologize for the confusion on the blinding. The behavioral analysis was carried blind of experimental group. This was not the case for the electrophysiology however.

9. Why is there a reduction in failures with CNO application in the Ctrl AAV-YFP group, and why does this reduction persist even after discontinuation of CNO. If this is due to learning how to escape the shock, why is this effect not seen in the Ctrl DREADD group? Why is there no effect of CNO on failure rate in the Ctrl DREADD condition?

The reviewer points to an important aspect we have noticed indeed. We do agree with the reviewer's interpretation that this might be due to learning in escaping the shock. This was however not consistent across groups as not present in the Control DREADD animals. We feel however this point goes beyond the scope of the paper that more points at the differences with the MD group. This opens nevertheless new questions on the role of the LHb in associative learning, which are so far elusive and are among the research questions in our laboratory.

10. Supp fig3d demonstrates CNO application leads to a reduction in firing rate in the Ctrl DREADD condition. Why is this reduction in firing rate not correlated with a reduction in failures in the shuttle box experiments?

This is an interesting issue. If the relationship LHb firing and behavior was linear we would predict that reducing firing in Control mice would also further ameliorate the shuttling of animals during the test. One possible explanation is that this relationship is not linear, and that in average, animals cannot perform in this test better than what they do, possibly because of a "ceiling" behavioral performance. It would be interesting for future work to understand to which extent LHb firing in physiological condition dynamically adjust behavioral performances.

11. Why is the CNO mediated current lessened in the MD animals?

DREADD receptors are likely hijacking the GIRK machinery to inhibit neuronal firing (Stachniak et al. 2014; Pascoli et al., 2016). This seems to apply also to the LHb. Considering that MS reduces GIRK expression, the CNO-driven GIRK current are smaller than in control. Yet this residual GIRK component is still efficient to modulate firing and behavior.

12. In sup Fig 3c if those recordings are done in control mice, a complementary experiment with MD mice should be shown. The efficacy of CNO to reduce the absolute firing rate should be assessed in MD animals.

The reviewer is right. In order to even better evaluate the effect of CNO on LHb function, we addressed this issue *in vivo* using single-unit recordings in anesthetized mice. We now report that CNO efficiently reduces activity in DREADD expressing animals, both control and MS. Importantly the CNO effect led hyperactivity in the MS mice to be restored.

13. Supp fig 3d shows the relative firing rates, can the authors include the absolute firing rates?

As suggested by the reviewer, we performed more recordings in anesthetized animals to assess the effect of CNO on the average firing rate of MS mice. We find that on average, MS produces LHb hyperactivity and that CNO rescued this effect. This new data set is now reported in Supplementary Figure 4e.

14. If I-CNO is decreased in MD animals compared to Ctrl, why do the authors observe the same time course and relative decrease in habenular firing rate in both experimental groups after CNO application?

The point raised by the reviewer is indeed important. The CNO concentration is high and saturating therefore it will likely activate the total GIRK channels through the hM4 receptors. Despite GIRK reduction, their total activation is nevertheless efficient in reducing activity to a similar extent. Another alternative scenario is that CNO-dependent hM4 activation leads to decreased neuronal activity via GIRK-independent mechanisms that remains so far unknown, but that cannot be ruled out.

15. The cartoon schematic and histology in Fig 2f indicates the the DBS electrodes were implanted unilaterally, however the methods mention bilateral placement of DBS electrodes. Can the authors clarify?

We could not find the statement of bilateral implantation to which the reviewer referred to. Nevertheless, we now made sure again that in the methods, section DBS, unilateral implantation is stated.

16. The use of DBS in the lateral habenula to reduce symptoms of depression has been successfully attempted before in both animals and human patients. Can the authors compare their DBS protocol with those previously used, and place their manuscript in context with the previous literature?

The reviewer is right. We did attempt in our manuscript to contextualize our results in light of the published literature. Indeed the DBS protocol is the same used in the Li et al., 2011 article from the Malinow group that we cited in the discussion. We modify the text, and hope now that this section is clearer.

17. The author's DBS protocol appears to reduce excitatory signaling through a presynaptic mechanism, leading to a reduction in release probability. Can the authors please comment more on how they believe DBS is leading to decreased habenular activity. What is the cause of the observed decrease input resistance?

We thank the reviewer to raise mechanistic question that are as well important for us. The extent of stimulation is likely to mediate an aberrant calcium presynaptic homeostasis affecting vesicle release. This mechanism may occur at the neuronal and non-neuronal level (Gradinaru et al., 2009). We also report that DBS also hyperpolarizes neurons. This remains instead mechanistically more obscure. Some reports indicate that such DBS stimulation can engage potassium conductances, as well as increase inhibitory tone. A more thorough investigation of such mechanisms

will be required to understand the various processes engaged by DBS, however we hope the reviewer agrees that it goes beyond the scope of this work.

18. Why wasn't the one hour protocol used in the initial characterization? I believe this is important as the effects on neuronal activity of 5 min vs 1 hour of DBS could be different.

We do agree with the reviewer that this is an important point. In a new set of data performed in anesthetized mice, we delivered a DBS protocol for 1 hour, and observed a similar reduction in activity compared to the five minutes stimulation. These data are now included in the revised Figure S4.

Reviewers' comments:

Reviewer #2 (Remarks to the Author):

While I appreciate the fact that the habenula is being attenuated and not shut down, the possibility of an offsetting effect remains (the authors should address the Moreines et al. Neuropsychopharmacology 2016 manuscript that argues quite convincingly that the habenula is not a primary driver in depression). The experiments as described fail to remove this possibility, and the discussion as it stands can be quite misleading.

Reviewer #3 (Remarks to the Author):

The authors have largely addressed the issues raised. There is still an issue regarding novelty, given that this group has published a very similar result, using learned helplessness (Lecca et al, 2016) rather than MD. They do reverse the behavioral phenotype in this study with DREADS and DBS.

With respect to an issue raised by reviewer 1, who suggests that the current study is not consistent with the Li et al, 2011; and the response from the authors is that their miniature recordings are not consistent with the Li et al, 2011 study:

It is important to note that NO conclusion can be made from a failure to detect a significant increase in mini frequency in the current study [i.e. this does not mean that there was no difference]. Indeed, the frequency of mEPSC in this study (Sup Fig 3a) do show an increase in the number of cells with high frequency (e.g. > 8Hz), which is EXACTLY what was seen in the Li et al paper. Had these authors recorded from more cells (the Li et al paper recorded from up to 85 cells per group rather than the 15-20 cells in this study) they may have succeeded in detecting a significant difference; note the data in this paper are clearly not normally distributed, and these authors used a t-test which is not appropriate. The only conclusion that can be drawn from this study in this area is that they likely did not record from a sufficient number of cells.

The authors also address a potential discrepancy with the Shabel et al, paper, with respect to the lack of effect of MD on the ratio of excitatory/inhibitory transmission. Again, this is unwarranted given that the Shabel et al paper recorded specifically from evoked EP inputs. This study addresses the ratio of excitatory/inhibitory transmission by looking at miniature EPSCs and IPSCs, which could have come from inputs originating from ANYWHERE.

Point by point response to Reviewers.

Reviewer #2

1. ... the authors should address the Moreines et al. Neuropsychopharmacology 2016 manuscript that argues quite convincingly that the habenula is not a primary driver in depression

The reviewer points to a very elegant study that demonstrates the importance of the IL-PFC-mVTA pathway in the effects produced by CMS on DA neurons firing. Together with our results, this study raises the interesting possibility that LHB may control MS-driven depressive-like states via downstream pathways different than the midbrain.

Accordingly to the reviewer suggestion, we now highlight in our discussion, not only the importance of LHB in depression (Li et al., 2011; Li et al., 2013; Seo et al., 2017; Lecca et al., 2016 and others), but also alternative important circuits like the one proposed in the Moreines work.

Reviewer #3

1. The authors have largely addressed the issues raised. There is still an issue regarding novelty...

We thank the reviewer for acknowledging our work on the resubmission. Regarding novelty, we still believe this work covers many aspects that were elusive in the previously published work (Lecca et al. 2016). For instance, the cellular basis of MD, the synaptic relevance of GABA-B receptors in MD, and the causality between LHB excitability, and depressive phenotypes.

2. It is important to note that NO conclusion can be made from a failure to detect a significant increase in mini frequency in the current study

We acknowledge the reviewer's point. We have now tone down in our discussion the overstatement that MS does not affect mEPSCs. We now leave open the possibility that such adaptation could also occur in MS mice likely in a circuit-specific manner, as it was demonstrated by Li et al., 2011.

3. The authors also address a potential discrepancy with the Shabel et al, paper, with respect to the lack of effect of MD on the ratio of excitatory/inhibitory transmission. Again, this is unwarranted given that the Shabel et al paper recorded specifically from evoked EP inputs. This study addresses the ratio of excitatory/inhibitory transmission by looking at miniature EPSCs and IPSCs, which could have come from inputs originating from ANYWHERE.

We would like firstly to apologize for the confusion. The way we wrote our sentence was misleading. We modified the text, and stated that we independently looked at mEPSCs OR mIPSCs (but not E/I ratios).

We nonetheless completely agree with the reviewer's point. Indeed our discussion already contained a sentence stating that mIPSCs can originate from several inputs, and that therefore the MS model could lead to input-specific modifications yet to be assessed.

As correctly pointed by the reviewer, we also re-analyzed the non-normally distributed data set using the more appropriate Kolmogorov-smirnov test, although the overall result remained unchanged.

REVIEWERS' COMMENTS:

Reviewer #2 (Remarks to the Author):

The authors addressed all issues.

Reviewer #3 (Remarks to the Author):

Nice paper.